# SARSPLEX: Multiplex Serological ELISA with a Holistic Approach

**DOI:** 10.3390/v14122593

**Published:** 2022-11-22

**Authors:** Kunal Garg, Sara Campolonghi, Armin Schwarzbach, Maria Luisa Garcia Alonso, Fausto M. Villavicencio-Aguilar, Liria M. Fajardo-Yamamoto, Leona Gilbert

**Affiliations:** 1Tezted Ltd., Mattilaniemi 6-8, 40100 Jyväskylä, Finland; 2ArminLabs, Zirbelstrabe 58, 86154 Augsburg, Germany; 3Estudios Analíticos Aplicados A La Clínica, Avenida Nuevo Mundo 11, Local 2, Boadilla del Monte, 28660 Madrid, Spain; 4Sanoviv Medical Institute, KM 39 Carretera Libre Tijuana-Ensenada s/n Interior 6, Playas de Rosarito 22712, Mexico

**Keywords:** COVID-19, SARS-CoV-2, novel coronavirus, multiplex, serological diagnostics

## Abstract

Currently, there are over 602 million severe acute respiratory syndrome coronavirus 2 (SARS-CoV-2) cases and 6.4 million COVID-19 disease-related deaths worldwide. With ambitious vaccine strategies, reliable and accurate serological testing is needed to monitor the dynamics of the novel coronavirus pandemic and community immunity. We set out to improve serological testing of the immune response against SARS-CoV-2. We hypothesize that by multiplexing the serological diagnostic test kit (SARSPLEX) and screening for three antibodies, an even more robust diagnostic can be developed. A total of 293 sera were analyzed for IgM, IgG, or IgA immune reactions to the subunit 1 spike glycoprotein and the nucleocapsid protein in a standardized ELISA platform. Testing IgM, IgG, and IgA demonstrated high positive and negative agreements compared to RT-PCR and serology reference tests. Comparison with the pre-2019-CoV (*n* = 102) samples highlighted the specificity of this test kit and indicated that no unspecific binding, even with the summer flu patients (*n* = 44), was detected. In addition, SARSPLEX demonstrated to be a valuable occupational surveillance tool used in a functional medicine facility. With increased and broader testing, SARSPLEX will be a valuable tool in monitoring immunity and aid in prioritizing access to the SARS-CoV-2 vaccine for high-risk patients.

## 1. Introduction

The novel 2019 coronavirus called COVID-19 or SARS-CoV-2 (severe acute respiratory syndrome coronavirus-2) originated in Wuhan, China, and is responsible for more than 6.3 million deaths globally as of 3 June 2022 [1,2,3]. In this scenario, there is an urgent need to provide novel diagnostics, drugs, and vaccines. Academia, industry, and governments worked closely together to tackle this pandemic, and various restrictions, including repeated social distancing, have been implemented to flatten the curve. To help improve the situation, we developed a multiplex diagnostic test kit for SARS-CoV-2 with improved analytical power.

The World Health Organization (WHO)’s message to all countries is “Test, Test, Test!” [4]. The asymptomatic carrier can transmit coronavirus within a communicable period of up to three weeks [5,6]. Thus, it is imperative to test asymptomatic individuals. The current diagnostic employed to test symptomatic people is the RT-PCR test to detect viral RNA. In general, RT-PCR test kits present many limitations [7], including long turnaround times with complicated and expensive operation. Interpretation of results requires skilled persons and certified laboratories, expensive equipment, and trained technicians; on average, €20.000 are needed to run the samples for only 90 patients; typically, 10–30% of positive cases are missed [8]. Nevertheless, serological testing is proving helpful in overcoming these pitfalls of RT-PCR [9,10].

Early profiling of immune responses to SARS-CoV-2 has demonstrated that IgM and IgG antibodies were both detectable on day 5 after the onset of symptoms and on day 14 for IgG [11]. In one study, researchers suggested the employment of serological testing as early as 3 days past the onset of symptoms [12]. Antibody testing can help provide epidemiological data on emerging coronavirus infection, reduce the prevalence of coronavirus and the magnitude of the epidemic/pandemic, and help determine the extent of community spread. By determining how many people in population groups have been exposed to SARS-CoV-2 (i.e., the true prevalence of infection), it is possible to improve exposure behavior and reduce the spread of infection. In addition, serological testing accurately identifies virus-spreading areas, directions, and the pandemic status. Similarly, serological testing can provide information on patient fitness and the effectiveness of treatment protocols.

For SARS-CoV-2, the spike glycoprotein and the nucleocapsid are gaining interest for serological tests. This is because SARS-CoV-2 (coronavirus) spike (S) glycoproteins promote entry into cells and are the main target of antibodies; S comprises two functional subunits responsible for binding to the host cell receptor (S1 subunit) and the fusion of the viral and cellular membranes (S2 subunit) [13]. SARS-CoV-2 is genetically related and presents similar epidemiology to SARS-CoV-1, as we learned from the SARS-CoV-1 2002–2004 outbreak [14,15,16].

In this outbreak, a highly restricted, immunoglobulin G-dominated antibody response was observed in patients with SARS, directed most frequently [89% by enzyme-linked immunosorbent assay (ELISA)] and predominantly at the nucleocapsid [17]. In this study, almost all the subjects without SARS had no anti-nucleocapsid antibodies, and the spike protein was the next most frequently targeted protein; however, only 63% of the patients (by ELISA) had an immune response to this protein. In addition, to date, no mutations of residues of SARS-CoV-2 S1 and S2 glycoproteins predicted to contact hACE2 (human receptor) have been observed among SARS-CoV-2 S sequences [13].

In the current context of diagnostic tests accessible in the EU, there are approximately 467 COVID-19 medical devices on the market or in development, and only 192 are CE-IVD marked [8,18]. Out of these 192 devices, 78 are PCR, 95 are immunoassay, and 19 are antigen-based. These 95 immunoassay test kits are based on using the spike (S) protein or nucleocapsid (N) alone or the S and N proteins in combination to screen primarily for IgM and IgG immune responses. Additionally, many test kits do not disclose the antigens used in their assay. Out of 95 immunoassays, only 16 utilize traditional EIA (enzyme-immunoassay) technology, with the remainder 79 tests employing rapid test format. Three kits use the S and N proteins together [17,19,20], but they only test for IgM and IgG. Lastly, there is only one diagnostic kit out of 16 ELISA-based tests that screen for IgM, IgG, and IgA; however, this screen is only against the N protein [11], and unfortunately, the sensitivity and specificity were not disclosed. As illustrated above and in the European Commission’s proposed guidelines [8], most current tests focus on testing one antigen and antibody at a time with sensitivities and specificities for IgM and IgG at 44.8–93%, and 82–99.39%, respectively. As for IgA, one device claims 92.7% sensitivity while using the N protein [8]. In addition, the evidence suggests that tests detecting two antibody types simultaneously (IgG and IgM) are superior to the ones testing for only one antibody [7,8,21].

Data from the present study suggests that targeting COVID-19 spike proteins and nuclear capsid proteins for IgM, IgG, and IgA antibodies can provide a holistic, more effective approach to screening people’s immune responses. Herein, we describe the SARSPLEX diagnostic test following Standards for Reporting of Diagnostic Accuracy Studies (STARD) and the European Commission’s proposed guidelines [8,22] with an emphasis on increasing the credibility and robustness of the test kit.

## 2. Materials and Methods

### 2.1. Patient Specimens

Anonymized, disregarded, and donated samples were retrospectively obtained from healthcare units to screen for infectious diseases according to their ISO15189 accreditation, the principles outlined in the Declaration of Helsinki, and the Finnish Act on the Use of Human Organs, Tissues and Cells 2001/101 [23]. In addition, reference samples were purchased from Discovery Life Sciences, and SARSPLEX (index test) was independently validated by clinical laboratories in Germany (ArminLabs), Spain (Estudios Analiticos Aplicados a la Clinica S.L., and Life Length), Sweden, Finland, and Mexico (Sanoviv Medical Institute). All samples which were independently validated were blinded and evaluated at Tezted Ltd. by a single operator. Appendix A display the list of samples and external diagnoses used in the clinical validation study along with SARSPLEX results. A total of 337 sera samples were tested; these included: 180 SARS-CoV-2 characterized sera specimens; 2 Human Parvovirus B19 IgG positive reference samples; 44 summer flu (2018; pre-2019-CoV specimens) patient sera samples; and 102 healthy sera samples or pre-2019-CoV specimens from 2016. Table 1 breaks down the sample size considered explicitly for IgM, IgG, and IgA immune responses in the present study.

### 2.2. ELISA Procedure and Cut-Off Determination Using Receiver Operating Curve

Subunit 1 of the spike glycoprotein (S) and the nuclear capsid (N) of SARS-CoV-2 was purchased from The Native Antigen Company. A series of serial dilutions of the antigens from (1/400 to 1/102400) 0.25, 0.124, 0.062, 0.031, 0.015, 0.007, 0.001, and 0.0009 μg were used to titrate S/N and analyze detection along with a positive (COP4, COP5, COP6) and negative sera (TEZ1). An ELISA was set up according to the previously described procedure [24]. Sera samples at 1/100 dilution were used to demonstrate titration of S and N antigens. Additionally, SARSPLEX was established according to ISO13485 protocols already established [24,25], except with the following additions: Human IgA (Sigma) was used as a positive control and Human IgG (Sigma) as a negative control for the secondary antibody anti-human IgA (Abcam).

The receiver operating curve (ROC) was employed to determine IgM, IgG, and IgA cut-off [26,27]. The pivot table feature in Microsoft Excel facilitated in creation of the ROC curve, and computing sensitivity and specificity using optical density values from SARS-CoV-2 antibody positive, SARS-CoV-2 antibody negative, SARS-CoV-2 RT-PCR positive, SARS-CoV-2 RT-PCR negative summer flu, and healthy sera samples or pre-2019-CoV specimens. On the ROC curve, the data point closest to 1 on the y-axis [True positive rate (Sensitivity)] was selected as the cut-off as it offers the maximum sensitivity and specificity [26,27]. GraphPad Prism version 9.4.1 (458) was used to create a violin plot to illustrate the clinical outcome for IgM, IgG, and IgA specimens (Table 1) relative to the cut-off and borderline. Clinical outcome or optical density values (ODI) are obtained by dividing the IgM, IgG, or IgA optical density (OD) of a specimen with the corresponding cut-off value from the ROC. The ODI values for specimens in Appendix A demonstrated a wide range (from 0.1 to 8), which required condensing the said range using the Log of ODI to create violin plots for IgM, IgG, and IgA.

The following equations were used for diagnostic sensitivity and specificity [26,27], respectively: 100 × the number of true positive values (TP) divided by the sum of the number of true positive values (TP) plus the number of false negative values (FN); or 100 × TP/(TP + FN), and 100 × the number of true negative values (TN) divided by the sum of the number of true negative plus the number of false positive (FP) values, or 100 × TN/(TN + FP) [26,27]. Lastly, the area under the curve (AUC) and the Youden Index (J = sensitivity+specificity-1) helped to estimate the overall ability of IgM, IgG, and IgA assay to distinguish between healthy and diseased specimens [26,27]. AUC or J values range between 0 to 1, wherein values closer to 1 suggest that the ELISA can effectively discriminate between the healthy and diseased samples [26,27]. The division of samples for sensitivity and specificity analysis is provided in Appendix A.

### 2.3. Precision

Intra- and inter-day assays for precision were conducted by performing intra- and inter-day assays and analyzing the coefficient of variance percentage of each of these tests [28,29]. For the intra-day assay, four plates on one day were performed with two different operators. The inter-day assay included two plates with two operators, and a comparison of the OD values to the previous OD values from the intra-day assay was compared.

### 2.4. Clinical Performance

Clinical precision was analyzed in evaluating the closeness of agreement between independent test results of the index test (i.e., SARSPLEX) performance while using Cohen’s kappa analysis. Cohen’s kappa (k) is a statistical method to assess the reliability of positive and negative agreement observed between two diagnostic test results [30,31,32]. Cohen’s k ranges from −1 to +1 wherein k values ≤0 indicate no agreement, 0.01–0.20 as none to a slight agreement, 0.21–0.40 as fair agreement, 0.41–0.60 as moderate agreement, 0.61–0.80 as substantial agreement, and 0.81–1.00 as almost perfect agreement. Cohen’s k was calculated using GraphPad (https://www.graphpad.com/quickcalcs/kappa1/ accessed on 18 November 2022).

Positive and negative agreement were analyzed according to the following: positive agreement of positive immune response between index test and reference test divided by the total number of positive samples; and negative agreement of negative immune response between index test and reference test divided by the total number of negative samples [26]. MEDCALC^®^ diagnostic test evaluation calculator was used to assess the positive and negative predictive values (PPV, NPV) for SARSPLEX compared to RT-PCR or serology reference tests (https://www.medcalc.org/calc/diagnostic_test.php accessed on 18 November 2022). Reference tests refer to RT-PCR tests manufactured by Autoimmun Diagnostik GmBH and Mobidiag Oy or serology tests developed by Euroimmune, Ortho clinical diagnostics, Mindray, and more.

### 2.5. Occupational Surveillance

Sanoviv Medical Institute is a Functional Medicine facility located in Baja California, Mexico, with more than 170 active employees that assessed SARSPLEX as an occupational surveillance tool. When the WHO officially declared the COVID-19 pandemic, Sanoviv’s internal regulations were updated to face and prevent a possible breakout threat across the staff. Before access to the hospital, all employees were screened for (1) COVID-19-like symptoms, (2) vital signs (i.e., body temperature and oxygen saturation), and (3) exposure to someone with flu-like symptoms in the past three days. If all three aspects were reported as negative or normal, then employees could enter the hospital wearing a mask and were advised to wash their hands when needed. If a team member reported any symptoms, exposure to someone with flu-like symptoms, or abnormal vital signs, they had to quarantine for 14 days, and a confirmatory COVID-19 test (RT-PCR) was indicated (Appendix A). In this setting, SARSPLEX was used to evaluate serology across time in Sanoviv staff (Appendix A) and to guide triage updates.

## 3. Results

With respect to STARD parameters samples [22] and EC’s proposed guidelines [8], antigen titration limit, specificity, sensitivity, efficiency, robustness, precision, and selectivity are described below.

The SARS-CoV-2 antigen titration for IgM, IgG, and IgA immune responses are portrayed in Figure 1, Figure 2 and Figure 3. The positive immune responses in Figure 1, Figure 2 and Figure 3 against the SARS-CoV-2 antigens have serially decreased, as expected in this serially diluted experiment. The negative reference sera demonstrated consistent basal reaction near zero OD. The IgM, IgG, and IgA immune responses with positive and negative serum samples converged when 15 ng of antigen was coated in the wells. Thus, the SARS-CoV-2 antigen titration limit for all three immune responses (IgM, IgG, and IgA) of SARSPLEX is observed at 31 ng of total antigen per well.

ROC analysis helped establish the cut-off, AUC, sensitivity, specificity, and Youden index (J) for all three antibodies (Figure 4). The AUC and Youden index (J) values for all three antibodies ranged between 0.81 to 0.93 and 0.57 to 0.86, respectively, indicating that the assay can discriminate between healthy and disease specimens (Figure 4). Sensitivity and specificity analysis for IgM immune response was evaluated at 70% and 94%, respectively (Figure 4); for IgG, the clinical sensitivity and specificity were 93% (Figure 4); and for IgA, 63% and 94%, respectively (Figure 4).

Additionally, SARSPLEX demonstrated high precision with robust inter- and intra- day performance. The intra-day assay coefficient of variation for IgM, IgG, and IgA was noted as 2.4%, 9.6%, and 6.4%, respectively. Moreover, the coefficient of inter-day assay variation for IgM, IgG, and IgA were 2.1%, 11.6%, and 8.3%, respectively.

Screening of samples (Table 1) by SARSPLEX for IgM (*n* = 202), IgG (*n* = 293), and IgA (*n* = 139) immune responses to the S and N proteins of SARS-CoV-2 is provided in Figure 5, Figure 6 and Figure 7, respectively. The IgM cut-off (dotted red line) and borderline (dotted green line) were calculated using ROC analysis (Figure 4A). Appendix A demonstrates line data for each specimen used in Figure 5 for IgM clinical validation. Within each violin plot (Figure 4), the black dashed line represents the median distribution of Log (ODI) values in a particular specimen category. Likewise, the dotted line below and above the median denote 25% and 75% quartiles, respectively (Figure 5). For example, the median distribution for SARS-CoV-2 IgM and RT-PCR (≥19 days onset) positive specimens is above the IgM cut-off, but the median distribution is below the cut-off for SARS-CoV-2 IgM negative, pre-2019-CoV, and summer flu patient groups. Appendix A demonstrates line data for each specimen used in Figure 5 for IgM clinical validation.

For IgG immune responses (Figure 6), IgG cut-off (dotted red line) and borderline (dotted green line) were calculated using ROC analysis (Figure 4B). Differences in a head-to-head comparison with other SARS-CoV-2 serology tests using only spike 1 protein or the nucleocapsid protein were noted. For example, COP7 and COP8, previously IgG negative for SARS-CoV-2 S protein, were also below the IgG cut-off (Appendix A). Similarly, COP1, COP2, COP3, COP4, COP5, COP9, COP10, and COP12 were previously IgG positive for SARS-CoV-2 S protein and were also above the IgG cut-off for SARSPLEX (Appendix A). However, the COP6 specimen previously IgG negative for SARS-CoV-2 S protein was positive on SARSPLEX (Appendix A). The positive immune response seen with COP6 may indicate that the inclusion of the SARS-CoV-2 nucleocapsid protein is increasing the IgG sensitivity with SARSPLEX. In contrast to SARS-CoV-2 S-only ELISAs, Okba and colleagues [33] have demonstrated that their in-house ELISA with SARS-CoV-2 nucleocapsid protein showed better sensitivity and specificity. In addition, Leung and colleagues [17] have also indicated that SARS-CoV-1 nucleocapsid ELISA is more sensitive than the full-length spike protein ELISA.

Lastly, Figure 7 presents the IgA immune responses to the proteins of the S and N of SARS-CoV-2 in SARSPLEX. IgA cut-off mark was established using the ROC (Figure 4C). Similarly to Figure 6, certain dissimilarities in a head-to-head comparison with other SARS-CoV-2 serology tests using only spike 1 protein or the nucleocapsid protein were noted. For example, COP8, previously IgA negative for SARS-CoV-2 S protein, was also seen below the IgA cut-off (Appendix A). Similarly, COP2, COP3, COP4, COP5, COP6, COP7, COP10, COP11, and COP12, were previously IgA positive for SARS-CoV-2 S protein was also above the IgA cut-off (Appendix A). However, COP1 and COP9 specimens that were previously IgA positive for SARS-CoV-2 S protein were negative on SARSPLEX (Appendix A). COP1 and COP9 demonstrated OD values in line with pre-2019-CoV and summer flu patient groups, indicating that the said COP specimens may have been falsely diagnosed as positives. Euroimmune (reference test; www.euroimmun.com, p. 9. accessed 18 November 2022) instruction for use for product EI2606-9601A indicated that up to 13% of healthy donors could react positively to the full length of SARS-CoV-2 S protein.

Clinical performance was also evaluated by analyzing the positive and negative agreement with the reference tests (i.e., RT-PCR & serology) and the index test (i.e., SARSPLEX) (Table 2 and Table 3). The SARS-CoV-2 commercial RT-PCR comparator or reference tests manufactured by Autoimmun Diagnostik GmBH and Mobidiag Oy were utilized (Table 2). Moreover, head-to-head testing with SARS-CoV-2 commercial antibody tests from manufacturers such as Euroimmune, Ortho Clinical Diagnostics, Mindray, and others was performed (Table 3). The number of positive/negative specimens in the reference test and the corresponding number of specimens with positive or negative immune responses on the index test are outlined in Table 2 and Table 3.

Substantial positive agreements are seen with IgM (78%), IgG (73%), and IgA (81%) when SARSPLEX is compared to RT-PCR results with disease onset greater than 19 days (Table 2). For negative agreements, IgM (91%), IgG (93%), and IgA (92%) demonstrated substantial agreements according to Cohen’s kappa interpretation [k ≥ 0.61–0.80 (Table 2]. The PPV and NPV percentages ranged between 63% to 81% and 92% to 95%, respectively (Table 2). Likewise, the false positive and false negative rates ranged between 7% to 9% and 19% to 27%, respectively (Table 2). The false positive and negative rates may be high due to the different proteins we have in our test kit (S1 and N compared to their full-length S protein), the 25% error rate found for RT-PCR tests, or analysis with a small sample size. A more extensive study will allow for an improvement in this percentage.

Table 3 demonstrates a head-to-head comparison of clinical outcomes between serological reference tests and the index test (Table 3). Following Cohen’s k analysis (Table 3), the positive and negative agreements for IgM and IgG were between 0.60 to 0.84 (i.e., moderate to almost perfect agreement with reference tests). In the case of IgA (Table 3), the clinical agreement with reference tests is moderate (k = 0.60). In particular, the positive and negative agreements for the three antibodies ranged between 65% to 93% and 94% to 100%, respectively (Table 3). In the case of PPV and NPV, the percentages ranged between 13% to 100% and 72% to 95%, respectively (Table 3). The false positive and false negative rates in Table 3 can be seen as low as 0% and 8%, respectively, most likely because the index test (Table 3) is compared to a serological test and not an RT-PCR test (Table 2). Borderline results for all specimens in Table 2 and Table 3 except COP 141 (Appendix A) were considered negative because the optical density value for COP 141 was 0.999.

In addition to clinical performance assessments in Table 2 and Table 3, between May 2020 and January 2021, SARSPLEX was used in Sanoviv Medical Institute as part of their Staff COVID-19-Surveillance Protocol. IgA, IgM, and IgG serology status among 170 employees were tracked (Figure 8) to identify potential exposure to COVID-19 and to determine if there was an urgent need to establish close contact follow-up between different departments. Due to the attendance of immunocompromised patients, a confirmatory test (RT-PCR) was indicated in asymptomatic employees who resulted positive for either IgM or IgA antibodies or both (borderline results were considered positive). Even though SARSPLEX was not initially implemented for detecting cases, it was useful in identifying 23 asymptomatic COVID-19 RT-PCR positive employees (Figure 8, Appendix A). Moreover, departments having a team member with positive or borderline antibodies (either IgA or IgM) were specifically follow-up by the Triage Department.

## 4. Discussion

More testing of symptomatic and asymptomatic COVID-19 cases by RT-PCR and serology are needed globally. Serological tests for immune responses against the antigens of SARS-CoV-2 will provide real-time epidemiological data, improve the picture of the prevalence in the population, and determine the extent of community spread. SARSPLEX was developed as a multiplex diagnostic tool to test for the immune responses against the S and N proteins of SARS-CoV-2. While there is a consistent development of serological tests that focus on one antibody and one antigen at a time, reports show that testing for two antibody types at the same time (i.e., IgG and IgM) increases the overall sensitivity and specificity of the diagnostic tests compared to tools that test for only one antibody [7,8,21,33].

SARSPLEX demonstrated high analytical and clinical performance of sensitivity and specificity compared to other test kits when the multiplex nature of the test kit is observed (Figure 4, Figure 5, Figure 6 and Figure 7; Table 2 and Table 3 [7]). Additionally, SARSPLEX demonstrated high precision at a 95% confidence interval with robust inter- and intra-day performance. Lastly, SARSPLEX showed substantial agreement for all three antibodies compared to commercial COVID-19 RT-PCR (Table 2) and serology (Table 3) reference tests. Overall, screening for all three antibodies improved SARSPLEX clinical performance and utility. The clinical agreement seen with the comparison results for RT-PCR (Table 2) may be due to the inherent RT-PCR error rate of 25% or delay in seroconversion [37].

Cross-reactivity between SARS-CoV-1 and SARS-CoV-2 is possible because of the similar structural proteins (full-length spike protein) that they share [38]. Here, we valued the use of recombinant proteins and subunits of the spike protein to lessen the potential cross-reactivity. More importantly, immune responses to SARS-CoV-2 or SARS-CoV-1 have been characterized in the general public, as their viral epitopes of the full length of S and N proteins may cross-react with human specimens that have encountered parainfluenza viruses, adenoviruses, H1N1, H7N9 influenza A, and influenza B strains, and common human coronaviruses strains such as 229E, OC43, and NL63 [39,40,41]. This immune response to the S protein seen in these individuals is most likely due to the conserved area of the S2 subunit of the virus [39,40,41]. Herein, we used the subunit S1 of the spike protein to increase the selectivity of the test kit and thereby inherently decreasing the likelihood of cross-reactivity. In addition, both the general public (***n*** = 102: healthy) and the summer flu (*n* = 44: Pre-2019-CoV) groups did not demonstrate cross-reactivity with our test kit (Figure 4, Figure 5 and Figure 6). Only a slightly positive response from Human Parvovirus B19 reference samples to SARSPLEX proteins (Figure 4, Figure 5 and Figure 6) was observed; however, others have seen these responses previously [33]. Simultaneous antibody testing for parvovirus B19 and SARS-CoV-2 may help reduce COVID-19-related misdiagnosis due to false PCR positivity, as recently demonstrated by Şenol G and Demirci F [42].

Independent clinical validation of SARSPLEX was conducted in two independent laboratories with inherent reproducibility and robustness (Appendix A). SARSPLEX precision and robustness should be innate, but local variations due to the lack of SARS-CoV-2 reference sera material may occur. Due to this, Tez1 has been added to the test kit as a cutoff control and performance control. A larger cohort of patients in the thousands, with varying degrees of severity, and an increase in specimen samples (i.e., for parainfluenza viruses, adenoviruses, H1N1, and H7N9 influenza A and influenza B strains, and common human coronaviruses strains such as 229E, OC43, and NL63) will support the current findings and validate the sensitivity of SARSPLEX on a larger scale [33,43]. Similarly, in this validation, the sample size provided a 95% confidence level, similarly to the more extensive studies reporting on the same CE and IVD-marked test kits in Europe [7].

## 5. Conclusions

In conclusion, SARSPLEX was helpful as an occupational surveillance tool to evaluate worker community exposure to COVID-19 and prevent an outbreak that could potentially harm hospitaller operations.

## Figures and Tables

**Figure 1 viruses-14-02593-f001:**
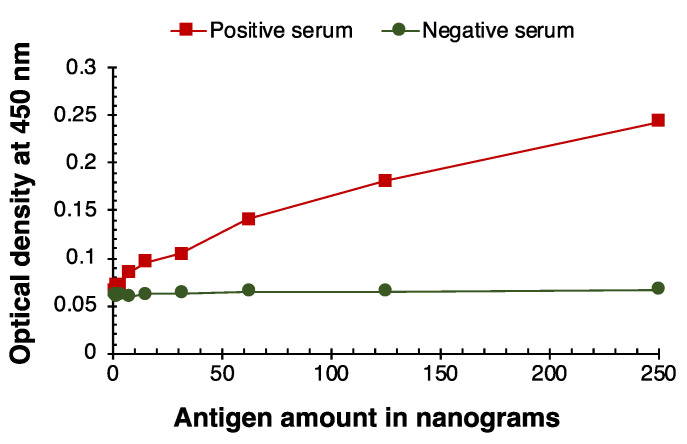
SARS-CoV-2 antigen titration analysis for IgM immune response. IgM immune responses by positive (COP6) and negative serum samples (Tez1) converged when 15 ng of antigen was coated in the wells. The SARS-CoV-2 antigen titration limit is observed at 31 ng of total antigen per well.

**Figure 2 viruses-14-02593-f002:**
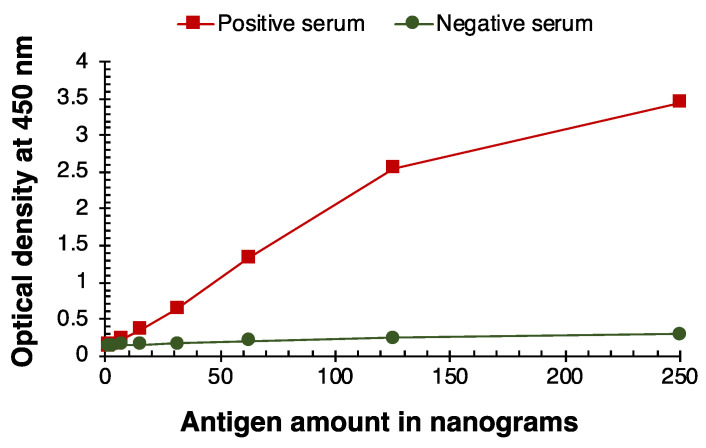
SARS-CoV-2 antigen titration analysis for IgG immune response. IgG immune responses by positive (COP5) and negative (Tez1) serum samples converged when 15 ng of antigen was coated in the wells. The SARS-CoV-2 antigen titration limit is observed at 31 ng of total antigen per well.

**Figure 3 viruses-14-02593-f003:**
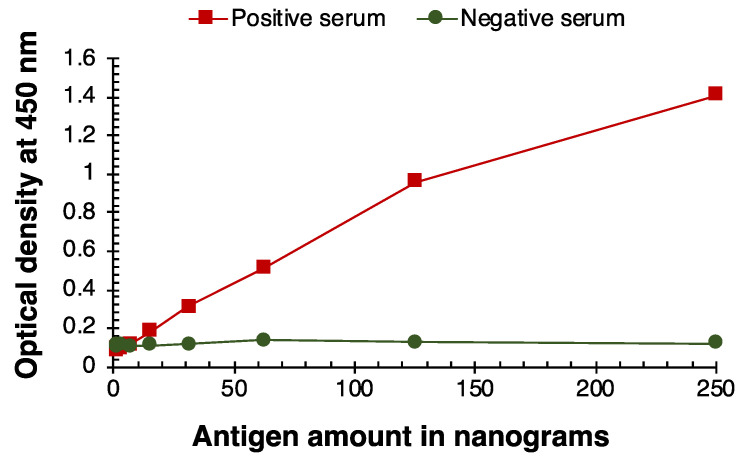
SARS-CoV-2 antigen titration analysis for IgA immune response. IgA immune responses by positive (COP4) and negative (Tez1) serum samples converged when 15 ng of antigen was coated in the wells. The SARS-CoV-2 antigen titration limit is observed at 31 ng of total antigen per well.

**Figure 4 viruses-14-02593-f004:**
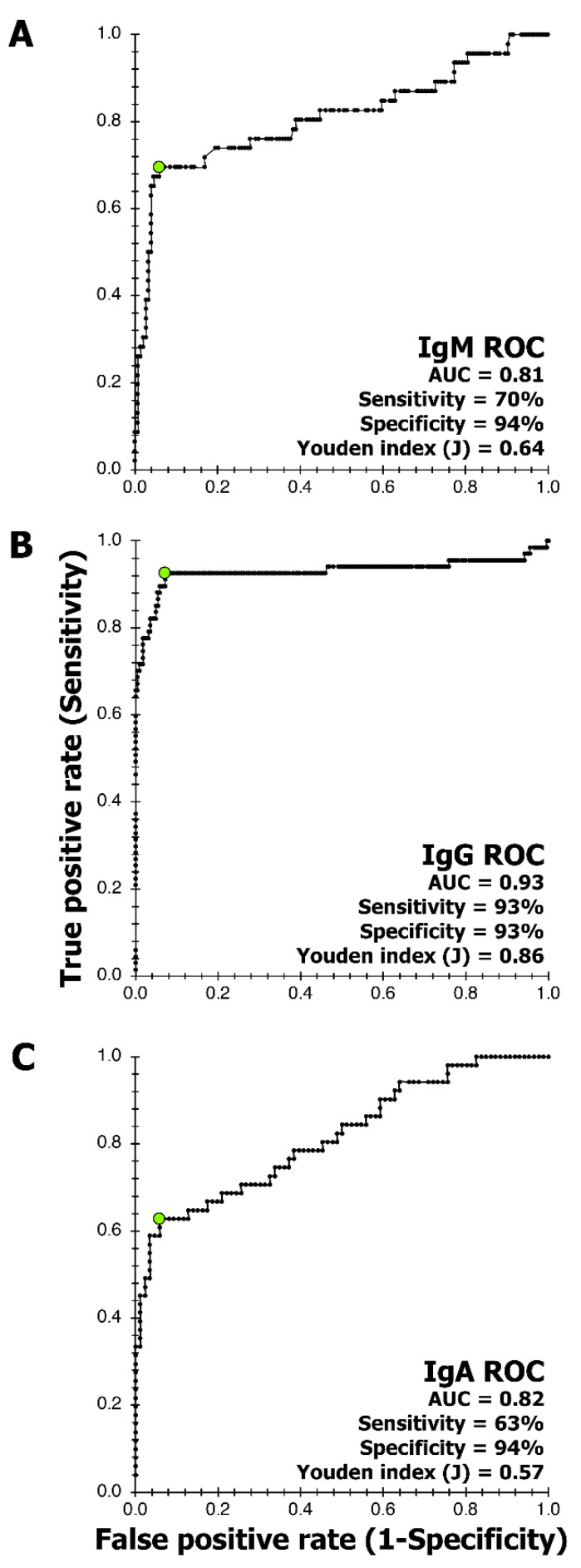
Receiver operating curve (ROC) analysis for IgM, IgG, and IgA, (**A**–**C**), respectively. The cut-off point for each antibody (green dot) on the ROC curve is closest to 1 on the y-axis [True positive rate (Sensitivity)] and offers the maximum sensitivity and specificity. The test kit’s clinical cut-off marks used to indicate positive and negative immune response performance was evaluated by ROC analysis using SARS-CoV-2 characterized sera, summer flu, and healthy sera samples or pre-2019-CoV specimens (Table 1 and Appendix A). Borderlines were 0.8–1 optical density index for IgM (**A**) and 0.9–1 for IgG (**B**) and IgA (**C**) immune responses [34,35,36]. AUC refers to the area under the curve. The pivot table feature in Excel facilitated in creation of the ROC curve, and computing sensitivity and specificity using optical density values from SARS-CoV-2 antibody positive, SARS-CoV-2 antibody negative, SARS-CoV-2 RT-PCR positive, SARS-CoV-2 RT-PCR negative summer flu, and healthy sera samples or pre-2019-CoV specimens.

**Figure 5 viruses-14-02593-f005:**
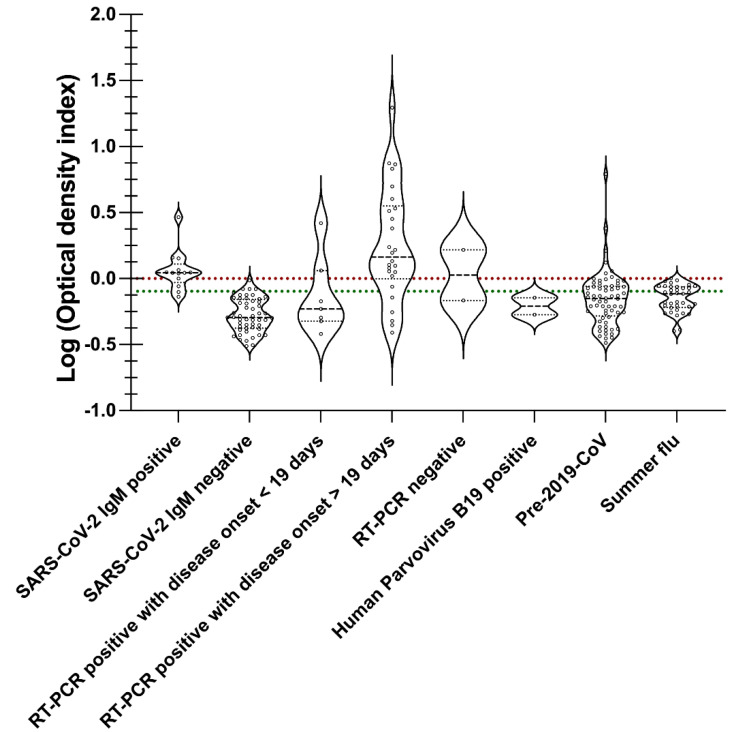
IgM immune responses to SARSPLEX S and N ELISA. IgM cut-off (dotted red line) and borderline (dotted green line) were calculated using ROC analysis. According to standard procedures, the Borderline was 0.8–1 optical density index (ODI) for IgG [34,35,36].

**Figure 6 viruses-14-02593-f006:**
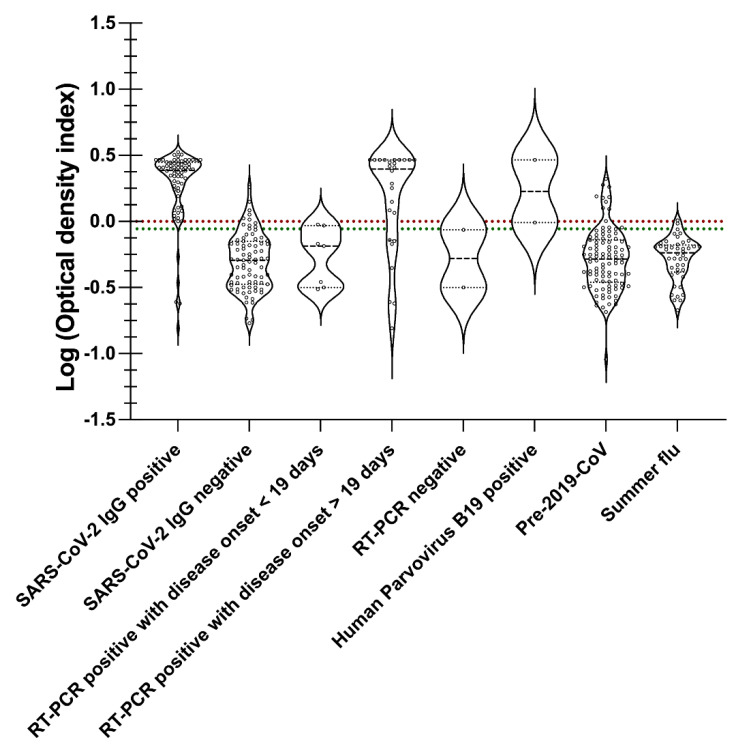
IgG immune response to SARSPLEX S and N ELISA. IgG cut-off (dotted red line) and borderline (dotted green line) were calculated using ROC analysis. According to standard procedures, the Borderline was 0.9–1 optical density index (ODI) for IgG [34,35,36].

**Figure 7 viruses-14-02593-f007:**
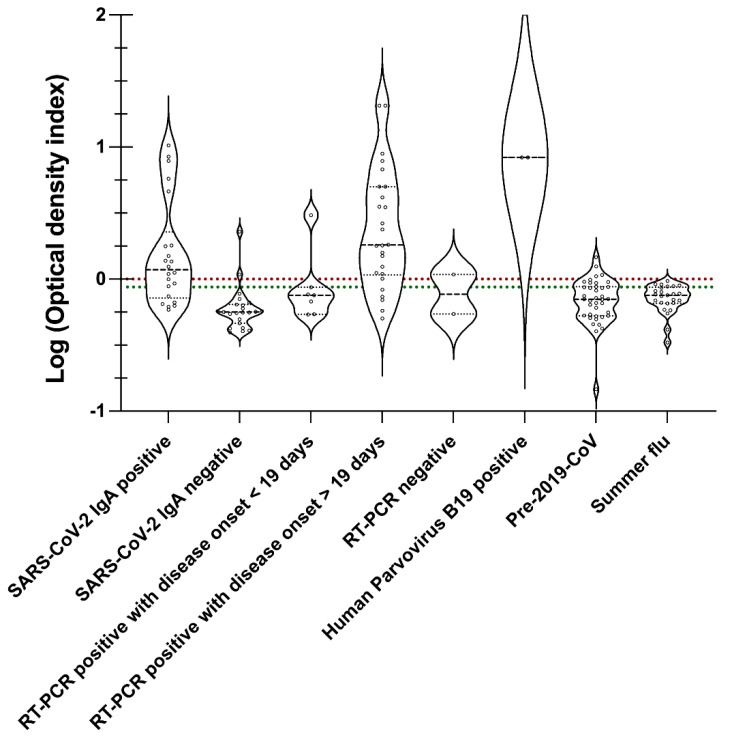
IgA immune response to SARSPLEX S and N ELISA. IgA cut-off (dotted red line) and borderline (dotted green line) were calculated using ROC analysis. According to standard procedures, the Borderline was 0.9–1 optical density index (ODI) for IgG [34,35,36].

**Figure 8 viruses-14-02593-f008:**
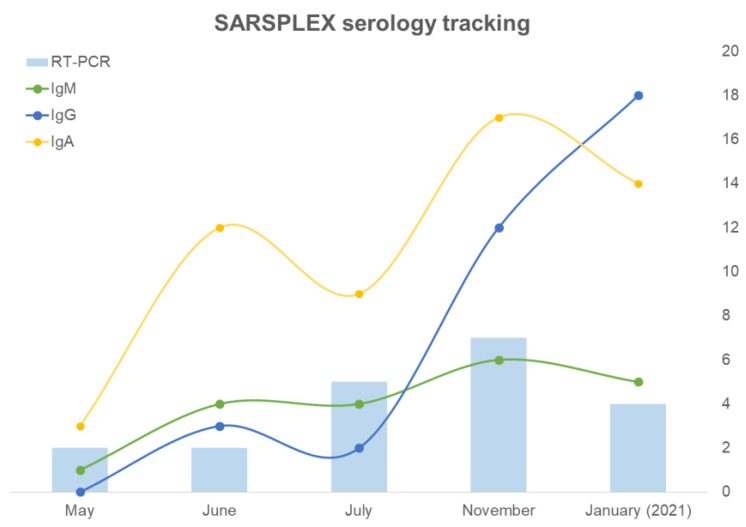
SARSPLEX serology levels across time in Sanoviv staff. Positive cases were confirmed by RT-PCR (Bars) and the data correlated with IgM and some IgG positives cases as was expected. The increased IgG and IgA antibodies in the last sampling revealed a normal immune system behavior. In this setting, SARSPLEX allowed the detection of COVID-19 cases and a follow-up of the immune system response in positive COVID-19 individuals and the general population. Samples correspond to the same individuals on each occasion, and only those (23 cases) who underwent RT-PCR are shown. The missing values (N = 170 employees) in the graph correspond to the entirely negative cases.

**Table 1 viruses-14-02593-t001:** List of human specimen categories selected for SARSPLEX clinical validation.

Specimen Categories	IgM (n)	IgG (n)	IgA (n)
SARS-CoV-2 antibody positive	13	66	23
SARS-CoV-2 antibody negative	52	78	22
RT-PCR positive with disease onset <19 days	7	7	7
RT-PCR positive with disease onset ≥19 days	26	26	26
RT-PCR negative	2	2	2
Human Parvovirus B19 positive	2	2	2
Pre-2019-CoV	64	102	39
Summer flu	36	44	25
Grand total	202	293	139

**Table 2 viruses-14-02593-t002:** Positive and negative agreement between index test and SARS-CoV-2 commercial RT-PCR comparator or reference tests manufactured by Autoimmun Diagnostik GmBH and Mobidiag Oy.

		IgM	IgG	IgA
Reference tests *	RT-PCR positive with disease onset ≥19 days (*n*)	26	26	26
	RT-PCR negative, pre-CoV-2019, & summer flu (*n*)	102	148	66
Index test (SARSPLEX)	Positive immune response to RT-PCR positive (*n*)	20	19	21
	Negative immune responses to RT-PCR negative, pre-CoV-2019, & summer flu (*n*)	93	137	61
Performance analysis parameters	Positive agreement (%)	78	73	81
	Negative agreement (%)	91	93	92
	Positive predictive value (%)	69	63	81
	Negative predictive value (%)	94	95	92
	False positive rate (%)	9	7	8
	False negative rate (%)	23	27	19
	Cohen’s Kappa (k)	0.65 ^SUA^	0.61 ^SUA^	0.73 ^SUA^

* SARS-CoV-2 commercial RT-PCR comparator or reference tests manufactured by Autoimmun Diagnostik GmBH and Mobidiag Oy were utilized. ^SUA^ Cohen’s Kappa interpretation (k ≥ 0.61–0.80) = substantial agreement.

**Table 3 viruses-14-02593-t003:** Positive and negative agreement between index test and SARS-CoV-2 commercial serology comparator or reference tests manufactured by Euroimmune, Ortho clinical diagnostics, Mindray, and more.

		IgM	IgG	IgA
Reference tests *	Positive sera (*n*)	13	66	23
	Negative sera (*n*)	52	78	22
Index test (SARSPLEX)	Positive immune response to positive sera (*n*)	10	61	15
	Negative immune responses negative sera (*n*)	52	73	21
Performance analysis parameters	Positive agreement (%)	77	93	65
	Negative agreement (%)	100	94	95
	Positive predictive value (%)	100	91	94
	Negative predictive value (%)	95	94	72
	False positive rate (%)	0	8	5
	False negative rate (%)	23	8	35
	Cohen’s Kappa (k)	0.84 ^APA^	0.84 ^APA^	0.60 ^MOA^

* SARS-CoV-2 commercial RT-PCR comparator or reference tests manufactured by Autoimmun Diagnostik GmBH and Mobidiag Oy were utilized. ^MOA^Cohen’s Kappa interpretation (k ≥ 0.41–0.60) = moderate agreement. ^APA^Cohen’s Kappa interpretation (k ≥ 0.81**–**1.00) = almost perfect agreement.

## Data Availability

All data from the study is found within the Tables.

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
