# Peer review of "SARSPLEX: Multiplex Serological ELISA with a Holistic Approach"

_viruses, 2022, doi:10.3390/v14122593_

Round 1
Reviewer 1 Report (Previous Reviewer 2)
The new version of the paper “SARSPLEX: Multiplex serological ELISA with a holistic approach” by Garg et al, has undergone significant improvements. Whereas most of my concerns have been properly addressed, there is still a considerable amount of data handling that requires attention before the paper can be recommended for publication.
Major points
1. My main concern is the inconsistent interpretation of borderline results. Whereas borderline results in most cases are interpreted as negatives, in Table 2, IgM and “Positive immune response to RT-PCR positive”, a borderline result is interpreted as positive. In Table 3, IgA and “Positive immune response to RT-PCR positive”, two borderline results are interpreted as positive. The authors should use consistent interpretation for each target and based on the general pattern of the data at hand, my suggestion is to consistently score all borderlines as negatives.
2. The data used for making the ROC-curves including total number of samples should be clearly indicated for each target. The term “SARS-CoV-2 characterized sera” is ambiguous since it is not clear whether this refers to samples characterized for Ig only or also includes the samples characterized by PCR. Although the former option gives fewer samples, the stringency should be higher. The relative low values for sensitivity of IgM and IgA (70 and 63 per cent, respectively) may indicate that a suboptimal validation cohort is actually used.
Minor points
1. The two parvovirus B19 samples are irrelevant for the study since they do not add any information on SARS-CoV-2 status and should thus be removed. In line with this, the two SARS-CoV-2 negative samples add limited value since immune status is unknown and could also be skipped.
2. Based on the number of samples mentioned in Table S1, the grand total for IgM and IgG in Table 1 should be 293 and 139, respectively, not 328 and 146. (Some samples in Table 1 may have been counted twice). If this affects the given numbers for grand total in the abstract and the main text, this should be corrected as well.
3. In the sentence “RT-PCR positive with disease onset > 19 days” found in Table 1 and in Table 2, the sign “larger than” should be changed “to equal or larger than”.
4. In Table S1, I found 66 IgG positives whereas “67” are indicated in Table 1. Please double check.
5. Since it is not obvious that the data in Figures 4-6 presents log-normal distributions, a conversion to linear plots should be considered.
6. If the ROC-data in Figure 7 is used for making Figures 4-6, a change of order may facilitate for the reader.
7. In the revised Table S1, the results for samples “POS2*”/IgM and “4638*”/IgG differ significantly from the data in the previous version and might require further attention.
Author Response
Please see the attachment.

Reviewer 2 Report (Previous Reviewer 1)
The work has been improved both in terms of experimental design and writing. It can be accepted in the present form
Author Response
Reviewer’s point #1
The work has been improved both in terms of experimental design and writing. It can be accepted in the present form
Author response #1
We thank the reviewer for helping us improve the quality of our manuscript. We have utilized Grammarly to improve the overall English language and minor spell checks following the feedback.
This manuscript is a resubmission of an earlier submission. The following is a list of the peer review reports and author responses from that submission.
Round 1
Reviewer 1 Report
This is an interesting and detailed work describing a novel immunological approach based on multiplexing the serological diagnostic test kit named SARSPLEX in order to detect three antibody classes, i.e., IgM, IgG, or IgA in human serum semples. Experiments were performed on n=339 sera, mostly SARS-CoV-2 positive, which were compared to n=44 sera from flu patients.
Main results indicate tha t SARSPLEX can be considered a valuable occupational surveillance tool used to evaluate SARS-CoV-2 serology in humans.
1. General important observation, besides sensitivity and specificity, Receiver-operating characteristic (ROC) curves calculating the areas under the curves (AUCs), positive and negative predictive values asl well as the Youden’s Index should also be evaluated in order to increase the robustness of the serological method described in the present study
2. General, figures 1-3 a logarithmic scale on y axis might most likely improve the figures, especially near low amounts of antigens
Line 14. Better “severe acute respiratory syndrome coronavirus 2 (SARS-CoV-2) “
Lines 51-79 Besides the already quoted references, these supporting references should also be included. https://www.mdpi.com/2076-2607/10/6/1193 and https://journals.asm.org/doi/10.1128/JCM.02160-20 Numerous different serological/antibody assays are described in detail in both reviews
Line 33 “coronavirus)” ---> “coronavirus-2)”
Line 40 citation?
Line 45 RT-PCR test are mainly used for detecting viral RNA. This information should be included
Line 72 it sohuld be IgG
Line 73 ELISA should be “enzyme-linked immunosorbent assay (ELISA)” when mentioned for the first time
General, the exact cut off determination method should be stated in the methods section. Moreover, a cut off of two times the standard deviation (reported in figure captions) might not be enough for excluding false results. Is it possible the cutoff point for Ig seropositivity is too low? Authors should consider three times the standard deviation of mean (mean + 3 SDs), for instance DOI:10.3389/fmicb.2021.789991. In addition, how many samples were considered for calculating the cut off? Different cut off determination approaches are reported in detail here DOI: 10.1371/journal.pntd.0007158
Line 113 this typo should be corrected “evaluated at Te?ted Oy by”
Lines 285-289 it seems that different font sizes have been used
Line 318 this typo should be corrected “size . A”
General, I discourage mentioning tables and figures in the discussion
Line 401 “SARS-CoV” should be “SARS-CoV-1”
Line 417 this subhead title can be removed for a better readability of the discussion
Reviewer 2 Report
In the current paper “SARSPLEX: Multiplex serological ELISA with a holistic approach” by Garg et al, a novel method is diagnosis of present and past exposure for SARS-Cov-2 is presented. Although the diagnostic performance appears to be adequate and the paper contains new data, it is not ready for publication and significant revisions are required. The best way the validate a new method is a head to head comparison of unselected, clinical samples against a defined gold standard. In the current paper, the comparison appears to be done against a mixture of selected positive and negative SARS-Cov-2 samples. In total, 65, 144 and 37 clinical samples for IgM, IgG and IgA, respectively. In addition, the authors used 100, 146 and 64 supposedly negative samples IgM, IgG and IgA, respectively. Although the specificity appears the quite satisfactory for all targets, one limitation of the study is the relatively low number clinical samples for IgM and IgG.
1. The method presented in the paper is in the Introduction advertised as tool for screening of symptomatic individuals. This is for conceptual and logistic reasons not likely. First, although IgM responses can sometimes be detected after 5 days, this will be insufficient to cover the early phase and the diagnostic sensitivity will be low. (This is also what is found in the samples that were PCR positive with disease onset < 19 days). Second, although Elisa procedures can be easily processed at al large scale, the requirement for plasma/serum makes the sampling for this option much more cumbersome than saliva/nasopharynx samples used for PCR and antigen.
2. The text in general and the Introduction and Discussion in particular is somewhat fluffy, loaded with jargon and buzzwords and should be rewritten and kept more stringent. For instance, the sentence on line 82-83 is to me incomprehensive. Overstatements that are not covered by the presented data should be avoided (e. g. “superior analytical and clinical performance”, line 387; “almost perfect”, line 389).
3. Fig 1-3 shows titration curves the three targets, not limit of detection. Limit of detection as determination of analytic sensitivity is normally presented as the extrapolated concentration where 95 % of the samples are scored as positive. If this analysis is not done, I suggest that the nomenclature is changed is something more appropriate.
4. For obvious reasons, the interpretation of borderline results is for notoriously difficult and there usefulness in validations of novel techniques of limited value. I thus suggest that the data presented in Figure 8 in this paper is omitted.
5. The serologic methods used for the reference tests is mentioned in Figure legends but should also be clearly indicated In M & M or Supplementary Table for each sample/sample cohort. If available, the time period when the pandemic samples were collected should be presented.
6. Figure 7 presenting clinical performance is the identical data presented in Table as positive/negative agreement. Since the latter is a more adequate description in this case, I suggest the authors keep this one. The confidence intervals could be added to the tables. Figure 7 could be omitted.
7. The algoritm for the combined results of IgM/IgG/IgM should be clearly stated or (alternatively) be omitted if its usage is not convincingly demonstrated. In six of seven tables it appears to be identical to IgG.
8. Given the uncertainty of the SARS-CoV-2 status, the EBV and mycoplasma samples are of limited value and should be omitted. For analysis of cross-reactivity, the test could be challenged by related coronaviruses but this might be beyond the scope of the present investigation.
9. In the Results section, COP-7 is presented is as negative for SARSPLEX IgA (line 250-252) but is in Table S3, Figure 6 and Table 5 and 7 presented as positive.
10. The explanation in the Discussion that the low agreement with PCR positive with disease onset > 19 days (line 396-398) should be due extremely poor performance of the PCR text (25 % error rate) seems unlikely unless additional information is presented. An alternative explanation that is at least as likely is that a significant fraction of the PCR positives have not yet seroconverted.
11. The line in Table 2-7 on “Agreement according to Coehn’s K” is not informative and should be omitted or replaced by an explanatory sentence in the Result or Discussion section. For example, the ranking of the K value for IgA in Table 6 of “1” is ranked is “Almost perfect” suggest that it could be even better, which is difficult to fully understand.
12. The separation of plasma and serum samples in Table 3-4 and 6-7 adds limited value and could be omitted.